# Dimensions Used in Instruments for QALY Calculation: A Systematic Review

**DOI:** 10.3390/ijerph18094428

**Published:** 2021-04-21

**Authors:** Moustapha Touré, Christian R. C. Kouakou, Thomas G. Poder

**Affiliations:** 1Department of Economics, Business School, Université de Sherbrooke, Sherbrooke, QC J1K 2R1, Canada; moustapha.toure@usherbrooke.ca (M.T.); christian.kouakou@usherbrooke.ca (C.R.C.K.); 2Centre de Recherche de l’IUSMM, CIUSSS de l’Est de L’île de Montréal, Montréal, QC H1N 3V2, Canada; 3Department of Management, Evaluation and Health Policy, School of Public Health, Université de Montréal, Montréal, QC H3N 1X9, Canada

**Keywords:** QALY, utility, impact, instrument development, economic assessment

## Abstract

Economic assessment is of utmost importance in the healthcare decision-making process. The quality-adjusted life-year (QALY) concept provides a rare opportunity to combine two crucial aspects of health, i.e., mortality and morbidity, into a single index to perform cost-utility comparison. Today, many tools are available to measure morbidity in terms of health-related quality of life (HRQoL) and a large literature describes how to use them. Knowing their characteristics and development process is a key point for elaborating, adapting, or selecting the most well-suited instrument for further needs. In this aim, we conducted a systematic review on instruments used for QALY calculation, and 46 studies were selected after searches in four databases: Medline EBSCO, Scopus, ScienceDirect, and PubMed. The search procedure was done to identify all relevant publications up to 18 June 2020. We mainly focused on the type of instrument developed (i.e., generic or specific), the number and the nature of dimensions and levels used, the elicitation method and the model selected to determine utility scores, and the instrument and algorithm validation methods. Results show that studies dealing with the development of specific instruments were mostly motivated by the inappropriateness of generic instruments in their field. For the dimensions’ and levels’ selection, item response theory, Rasch analysis, and literature review were mostly used. Dimensions and levels were validated by methods like the Loevinger H, the standardised response mean, or discussions with experts in the field. The time trade-off method was the most widely used elicitation method, followed by the visual analogue scale. Random effects regression models were frequently used in determining utility scores.

## 1. Introduction

In the face of growing demand for health services, public and private agencies are increasingly interested in knowing the relative cost-effectiveness of programs [1]. In this setting, the quality-adjusted life-year (QALY) concept has grown in popularity and is now used as a measure of benefit in the economic evaluation of health programs and technologies all around the world [2]. The principle of the QALY is to combine the duration (mortality) and quality (morbidity) of life into a single measure [3]. Such a combination allows comparisons between various intervention in health care [4]. If duration is simply estimated in a QALY calculation as the number of years lived in a given health state, quality of life is characterized by a utility value between 0 and 1, where 0 represents death and 1 represents perfect health. Many instruments to measure the Q in QALY have been developed, while some are generic, others are specific [5]. The purpose of all these instruments is to reflect respondents’ perceived health onto a utility continuum in the aim to capture the comparative effectiveness of healthcare intervention or programs [1,2,6,7].

To be usable in cost-utility studies, instruments must meet several essential criteria. The development of these instruments is done in several stages to ensure their reliability and validity. These steps, which are common to both generic and specific instruments, are generally described under three aspects: development, validation of psychometric properties, and measurement (i.e., valuation exercise to elicit health state values) [8,9]. In the aim to face further challenges and better assess programs that impact current or new fields, researchers and decision makers need adapted instruments. To develop, adapt, or select the appropriate instrument is thus necessary, and to do so, it is essential to master the different stages of the development process. There is a rich literature dealing with the development of various instruments for evaluation purposes. However, few studies describe, through a clear process, the development of preference-based instruments that can be used for cost-utility analysis [10]. The purpose of this systematic review was to analyze the different phases of the development of the instruments used in QALY determination in various countries. More specifically, it was to determine the dimensions and levels used in these instruments and to specify how these dimensions and their utility scores were obtained. To report this, the stages mentioned above were followed. By doing so, we provide a synthesis of the instruments’ development process based on existing literature that can benefit to the research community when developing or improving instruments for QALY calculation. Next sections present the methodology used for the systematic review, the results and the discussion.

## 2. Method

### 2.1. Search Strategy

The databases consulted were Medline EBSCO, Scopus, ScienceDirect (Elsevier), and PubMed. Grey literature searches were also conducted via Google Scholar and ResearchGate. The bibliographic references of the selected articles were used as a source to find other relevant studies. The keywords used in the different databases were ‘QALY’, ‘quality adjusted life year’, ‘instrument’, ‘multi-attribute’, and ‘utility’. Using Boolean operators, combinations were made to refine the results and get closer to the type of study requested. There was no restriction on the publication date and only publications in English or French were considered. Searches were conducted in English in the databases mentioned above. The search procedure was conducted up to 18 June 2020.

### 2.2. Selection of Studies

In accordance with our literature search protocol (i.e., an unpublished 2-page document in French to ensure consistency and reproducibility), the selection of studies was based on the following criteria: Studies published in French or English; studies describing the development of instruments for QALY calculation; and studies addressing the general population or specific patient groups.

Studies dealing with draft versions of instruments that have been subsequently modified and published, using an instrument for QALY calculation without a description of its dimensions and levels, using instruments that do not measure health utilities, and dealing with the paediatric population were not included.

The selection of studies was conducted in 2 steps. A group of 2 reviewers (M.T. and C.R.C.K.) first made the first selection after reading the titles and abstracts. The selected articles were then read in full and only those that met the inclusion criteria were selected. At each step, in case of disagreement between the 2 evaluators, the reason for this disagreement was submitted to a third evaluator (T.G.P.) who performed an arbitration. A Kappa coefficient was calculated in both steps. Data extraction was done by one evaluator (M.T.) and validated by another one (C.R.C.K. or T.G.P.).

### 2.3. Data Analysis

Data extraction was performed using a form structured around the instrument’s development process. Thus, the main information to be collected was related to the 3 aspects of instrument elaboration: development, validation, and measurement. Specifically, we were interested in the target population, the type of instrument developed, the number and nature of dimensions and levels, the elicitation method and model used in the determination of utility scores, and the methods used to validate the tool and the utility algorithm. As regards to the dimensions or attributes selected in the instruments, we grouped them into the three main areas of health described by the World Health Organization (WHO), i.e., physical, mental, and social. Additionally, when available, we collected information on the sociodemographic characteristics of the participants in the different stages, and the method used to recruit them. The analysis of the quality of the studies was done with the COnsensus-based Standards for the selection of health Measurement INstruments (COSMIN) grid [11].

## 3. Results

### 3.1. Selection of Studies

Following our search strategy, a total of 4270 studies were found. At the end of the filtering processes, 50 articles were fully read and 46 met the inclusion criteria. Figure 1 shows the PRISMA flowchart describing the selection process. At the first stage of the selection, 2740 publications were excluded and a kappa coefficient equal to 0.37 was obtained. In the second stage of the selection, four studies were excluded because they did not concern instrument’s development process for QALY calculation or were related to the development process of previous versions of an instrument used in QALY determination that is no longer in use. Consequently, and as mentioned in the Method section, only updated versions of instruments were considered. A kappa coefficient of 0.65 was found at this stage and a third evaluator had to intervene to decide between disagreements related to two studies. This review thus considered 46 studies dealing with the development of 48 preference-based instruments usable in QALY calculation using their own value set.

### 3.2. Characteristics of the Selected Studies

Of the 46 studies that met the inclusion criteria, 12 dealt with the development of generic instruments (12 generic instruments) and the remainder (*n* = 34) were about the development of instruments for specific health conditions (36 specific instruments). Countries of application of these studies were United Kingdom (*n* = 20), United States of America (*n* = 6), Australia (*n* = 5), Holland (*n* = 5), Canada (*n* = 3), Spain (*n* = 1), Finland (*n* = 1), England (*n* = 1), and South Korea (*n* = 1). The remaining studies were carried out simultaneously in several of the above-mentioned countries (*n* = 3). The specific instruments developed refer to a wide variety of areas related to social care and dependency (*n* = 7), neurological disorders (*n* = 6), respiratory problems (*n* = 4), cancer (*n* = 3), diabetes (*n* = 3), sexuality/fertility (*n* = 3), bladder (*n* = 2), menopause/flushing (*n* = 2), musculoskeletal disorders (*n* = 2), vision/glaucoma (*n* = 2), digestive function (*n* = 1), and prostate (*n* = 1). All studies were published between 1998 and 2020.

### 3.3. Instrument Development

The development of preference-based tools provide a mean of measuring health states preferences in a field where such instruments are non-existent or to overcome the problem of unsuitability of existing instruments (e.g., sensitivity problems, missing dimension) [10,12,13]. Instruments vary in their composition, length, and in what they intend to measure. Then, the scheme (study process and design, target population, etc.) followed in the development of an instrument often depends on the motivation behind its creation [14]. A distinction will therefore be made between specific and generic instruments. Specific instruments are more oriented towards specific conditions or diseases but do not allow comparison between the quality of life of patients with different diseases, whereas generic instruments can be used by any type of patient, regardless of their health profile, and allow comparison between different patients with different diseases [14]. Thus, to allow for a better allocation of available resources, various generic and specific instruments have been developed. In addition, it should be noted that some of the preference-based tools were originally developed in that purpose while others were pre-existing instruments that were modified to be preference-based. A total of 48 instruments make up this review, 25 of which are the result of improvements to existing instruments and 23 of which were developed de novo.

Less than a quarter of the studies included in this review concerned the development of generic instruments (*n* = 12). These instruments are the 15-dimensional (15D), the assessment of quality of life 7-dimension (AQoL-7D), the assessment of quality of life-8 (AQoL-8D), the computerized adaptive tool 5-dimension (CAT-5D-QOL), the clinical outcomes in routine evaluation 6-dimension (CORE-6D), the EuroQol 5-dimension (EQ-5D), the health utilities index (HUI2 and HUI3), the patient reported outcomes measurement information system-29 (PROMIS-29), the quality of well being self-administered (QWB-SA), the recovering quality of life utility index (ReQoL-UI), and the second version of the short-form 6-dimension (SF-6Dv2). Half of these studies (*n* = 6) describe the improvement of a pre-existing tool because of limitations noted in its use. This is the case for Hawthorne (2009) [10], Seiber et al. [15], and Richardson et al. [16] who dealt with the development of parsimonious tools from AQoL and QWB, respectively, which would satisfy the axioms of utility theory and would be able to overcome the limitations observed in those instruments. To do so, they suggested switching from original versions (AQoL, AQoL-6D, and QWB) to AQoL-7D, AQol-8D, and QWB-SA, respectively. Hawthorne [10] thus retained eight items through an iterative process of entering and removing the items proposed in the AQoL model. This process was repeated until all possible combinations of items were examined. Richardson et al. [16] proposed to increase the sensitivity of AQoL to sight-related difficulties and disabilities. Vision-related quality of life (VisQol) was thus added as a dimension to AQoL-6D. Seiber et al. [15] explained the implementation of the QWB-SA, derived from the quality of well being (QWB). The QWB-SA is a tool offering the same properties as the latter, while being less time consuming and easier to use.

In the case of Herdman et al. [12] and Brazier et al. [7], the concerns was more about the sensitivity of previous versions. Consequently, they introduced the EQ-5D-5L and SF-6Dv2, respectively. These authors wished to remedy the problems of the ‘ceiling effect’ and ‘floor effect’ from which the first versions of these instruments suffered (respectively the EQ-5D-3L and the SF-6Dv1). The main changes were provided in the nature of the severity levels in different dimensions, leading to an increased number of possible combinations from 243 to 3125 for EQ-5D-5L and from 18,000 to 18,750 for SF-6Dv2. For this purpose, a literature review on the response scales and interviews with native speakers of the different target languages and experts were conducted. In addition, the exploratory factor analysis (EFA), confirmatory factor analysis (CFA), and Rash’s analysis made it possible to retain the elements relevant to the new tools. These same techniques were used in the development of CORE-6D, PROMIS-29, and ReQoL-UI. Indeed, factor analyses are techniques that ensure a certain structural independence of the dimensions defined in the instruments by identifying the factors underlying the correlation patterns in a set of observed variables [17]. The Rasch analysis is a probabilistic method that models the probability of correct response conditional on the levels of difficulty of the question and the respondent’s ability [18]. Rasch analysis therefore allows, using measurement intervals, to evaluate different psychometric properties such as unidimensionality (the degree to which the tool measures the same aspect), targeting (the degree to which the instrument is appropriate for respondents in terms of difficulty), item severity (the order in which items are set up), and separation (the way in which items distinguish levels of functionality in different domains) [19].

Table 1 provides an overview of the dimensions and levels covered by the different generic instruments identified, while Table 2 identifies the different methods used in the different phases of the development of these generic instruments. In Table 1, the instrument that covered the most dimensions was the QWB-SA, followed by the AQoL-8D, the AQoL-7D, the PROMIS-29, and the 15D. The ReQoL-UI records the fewest dimensions, which can be misleading since it covers many subdimensions in mental health. All instruments record dimensions related to physical discomfort/pain and almost all instruments had dimensions related to mobility/ambulation. Only one instrument (CORE-6D) did not record dimensions on mobility/ambulation. Four instruments reported dimensions relative to eating/nutrition and autonomy/control/dependence. Seven instruments addressed the sadness/depression issue. Five and eight instruments had dimensions related to mental/cognitive function and anxiety/distress, respectively. Five instruments were interested in well-being/happiness/satisfaction. Fertility was only considered in HUI2 and sexual activity in 15D and QWB-SA. Dimensions related to terror/panic/fear and humiliation/shame were present in two instruments (CORE-6D and PROMIS-29). The number of levels per dimension varied between 2 (QWB-SA) and 11 (PROMIS-29). The number of possible combinations ranged from 3125 (EQ-5D-5L) to 3.55957 × 10^24^ (AQoL-8D).

Among the specific instruments that were developed (*n* = 36), their authors were mostly motivated by a problem of inadequacy of existing tools due to their lack of sensitivity or their psychometrically invalid nature in their field of interest (*n* = 23). Other instruments (*n* = 13) were simply developed because of the non-existence of a measurement tool or the fact that existing tools were not usable in economic evaluation because they were not based on individual preferences. Table 3 shows the dimensions and levels used in the various specific instruments.

Several studies (*n* = 18) specified that a literature review of old instruments and exchanges with professionals and/or patients helped in the selection of dimensions and levels. In addition to these resources, half (*n* = 18) of the studies stated that they used empirical methods such as the factor analysis, Rasch analysis, standard psychometric criteria, and differential item functioning (DIF) in the selection of the dimensions and levels shown in Table 3.

### 3.4. Psychometric Validation

Following the selection of the items to make up the instrument, it was subjected to qualitative and quantitative tests to ensure its reliability, consistency, and validity (internal and external) [26,27]. Messick [28] defines validity as an integrated evaluative judgment of the degree to which empirical evidence and theoretical evidence supports the adequacy and appropriateness of interpretations and actions based on test scores or any other mode of assessment.

Among the 12 generic instruments, the method used to test the validation of dimensions and levels was provided for 6 tools (see Table 2). Hawthorne [10] tested the unidimensionality of the descriptive system and the degree of homogeneity using item response theory (IRT) and Loevinger’s H coefficient, respectively. Indeed, IRT was first proposed in the field of psychometrics and is currently widely used in the fields of health and education. Its methodology significantly improves the accuracy and reliability of measurement instruments while providing significant reductions in the time and effort required for assessments [18].

From their side, Herdman et al. [12] asked participants to assess the interpretability and plausibility of the instrument. Using subsamples, Brazier et al. [7] and Seiber et al. [15] used the DIF and the test–retest, respectively. In addition, the latter tested the impact of the questionnaire administration method on the scores obtained. The DIF examines the relationship between the response to an item and a group characteristic (e.g., gender, race, and level of education). Thus, the question answered is whether or not the response to a question in an item is due to belonging to a group [29]. As for the test–retest, it allows to measure the reliability of the instrument by observing the constancy of the scores by measuring a stable characteristic at different periods [30]. Sintonen [20] stated that for its validation, the 15D was compared to other instruments such as the Nottingham Health Profile (NHP), the 20-Item Short-Form Health Survey (SF-20), and the EQ-5D. Only three studies provided information on the samples used for the validation of the different generic instruments [7,10,12].

Regarding the validation of specific instruments, less than the third of instruments (*n* = 10) provided their validation method (see Table 4). The two versions of the DHP (DHP3 and DHP5) were validated by collecting the opinions of professionals in the field after presenting them with the results of the item selection. The sensitivity of OAB-5D and EORTC-8D was tested using the standardised response mean (SRM) on random samples from the initial database and on an independent sample of patients. This method was used to measure sensitivity in patients with minimal change in health status between visits. It is obtained by dividing the change in mean score by the standard deviation of that change [31]. The validity of the ASCOT was tested by comparing it with other instruments such as the EQ-5D and the general health questionnaire (GHQ-12). This was done using the Chi-square test and the analysis of variance. A comparison with other instruments was also performed for the DUI and P-PBMSI using the Cohen criterion, Spearman’s correlation, and Pearson’s correlation. A patient group test–retest was used for the validation of the CAMPHOR, the menopause specific health quality of life questionnaire, and the RSUI to assess the reliability and validity of the construction of these instruments. Finally, the IIEF was validated following confirmation of the consistency of the ordinal structure of its dimensions.

### 3.5. Measuring Utility Scores

The final step in the process of creating a preference-based instrument is the measurement of individual preferences. This involves assigning a utility score to the different possible health states described by each instrument. To do this, a health preference survey is filled out by a sample of individuals [1,32,33]. Different elicitation methods are used to obtain the preferences of individuals [32,33,34]. A review of elicitation methods is available in the work by Fauteux and Poder [32]. Due to the often large number of possible combinations offered by an instrument, a subset of health states is frequently chosen to be assessed directly by participants, and the utility levels or scores of the remaining health states are then modelled and estimated from the results of the sample of health states chosen at baseline [16]. In this exercise to assess the selected health states, two thirds of the instruments used the preferences of individuals from the general population (*n* = 33) compared to less than one third that used patient preferences (*n* = 9). Only five instruments were valued by both parties. More than two thirds of the elicitations of the selected health states were made by direct interviews (*n* = 34), eight instruments were evaluated through remote methods (online survey and postal mail), and one study used a pre-existing database. Five studies did not provide information on the mode of elicitation. In addition, 85% of the studies provided information on the number of participants, and of these, 95% provided details on the characteristics of the participants (e.g., age, level of education). However, only 41% of the studies (*n* = 19) stated that the sample used was representative of the target population.

There is some diversity in the elicitation methods used in studies dealing with the development of generic instruments. Time trade-off (TTO) was the most used method (*n* = 4), followed by the visual analogue scale (VAS) (*n* = 3), standard gamble (SG) (*n* = 1), and discrete choice experiment (with duration) (DCEtto) (*n* = 2). One study used a hybrid method combining VAS and SG (i.e., for HUI2 and HUI3). The utility scores obtained for the selected health states through these different elicitation methods provided scores for all other possible combinations of health states using different models. The additive regression model was used for AQoL-8D, 15D, QWB-SA, and CORE-6D; the conditional logit for SF6-Dv2 and the multiplicative model for CAT-5D-QOL, HUI2, and 3, and AQoL-7D. The random effects model was used for ReQoL-UI and the relativity model for the PROMIS-29.

The models, once estimated, were validated to ensure the reliability of the results. Different possibilities allow the validation of the models. For example, the preferred model for the AQoL-8D was the one that had the closest utility scores to the original instrument (AQoL) and the highest degree of correlation with it. For the CAT-5D-QOL, a comparison of its scores with those of the HUI3 allowed one to select the best specification. For the SF6-Dv2, heterogeneity was tested and the 15D had its preferred model selected using correlation analyses with different samples. For AQoL-7D, the analysis of its ability to discriminate between the general population and patients allowed its model to be validated. The analysis of the specification of the different models used (significance of the coefficients, mean absolute error, root mean standard error, etc.) made it possible to validate the best model for CORE-6D and ReQoL-UI. For the PROMIS-29, the ability of the model to predict pair-specific probabilities in terms of least squared error was used to determine the best model.

**Table 4 ijerph-18-04428-t004:** Methods used for the development of specific instruments.

Instruments	Method of Choice of Dimensions and Levels	Validation Method	Elicitation Method	Model Used	References
**Social care and dependency**					
**Aberrant Behaviour Checklist Utility Index (ABC-UI)**	Factor and Rasch analyses, consultation with clinical experts	Not found	TTO	Maximum likelihood with random effects	[35]
**Adult Social Care Outcomes Toolkit (ASCOT)**	Literature review on old instruments; empirical analysis	Comparison with other measurement tools	TTO; DCE; BWS	Multinomial logit model	[9]
**Carer Quality of Life—7 Dimensions (CarerQol-7D)**	Review of existing instruments; Experts’ opinion.	Not found	DCE	Panel mixed multinomial parameter model including main and interaction effects (MMNL)	[36]
**Dependency 6 dimensions (DEP-6D)**	Non available	Not found	TTO	Random effects regression model	[37]
**Impact of Weight on Quality of Life—Lite (IWQOL-Lite)**	Non available	Not found	DCE	Random effects ordered probit	[38]
**Index of capability for older people (ICECAP-O)**	Iterative interviews until convergence	Not found	Best–worst scaling (BWS)	Conditional logistic regression	[39]
**Older Persons Utility Scale (OPUS)**	Consultation with individuals drawn from local authority senior and middle managers	Not found	DCE	Random effects probit model	[40]
**Neurological disorders**					
**Alzheimer’s disease (AD-5D)**	Factorial analysis; Rasch analysis	Not found	To be developed	To be developed	[41]
**Amyotrophic Lateral Sclerosis Utility Index (ALSUI)**	Non available	Not found	VAS; SG	Multiplicative model	[42]
**Cerebral palsy-specific 6 dimensions (CP-6D)**	Factorial analysis, Rasch analysis.	Not found	DCE with duration (DCEtto)	Conditional logit, mixed logit	[43]
**Epilepsy-specific preference-based measure (NEWQOL-6D)**	Exploratory factor analysis, Rasch and psychometric analyses, DIF	Not found	TTO	Generalized least squares regression	[8]
**Multiple Sclerosis Impact Scale 29 (MSIS-29)**	Rasch model, basic psychometric criteria, clinical expert opinion	Not found	TTO	Random effects model	[44]
**Prototype Preference-Based MS Index (P-PBMSI)**	Rasch analysis, threshold graph, WHO International Classification of Functioning, Disability and Health.	Comparison with other instruments; Cohen criterion; Spearman and Pearson correlations.	VAS	Simple linear regression	[45]
**Respiratory problems**					
**Asthma Quality of Life (AQL-5D)**	Non available	Not found	TTO	Fixed-effect model	[46]
**Cambridge Pulmonary Hypertension Outcome Review (CAMPHOR)**	Percent affirmation of items; logit location in Rasch analysis	Test-retest	TTO	Ordinary least squares; Random effects model.	[47]
**Chronic obstructive pulmonary disease (COPD)**	Non available	Not found	TTO; VAS	Linear mix model	[48]
**Rhinitis Symptom Utility Index (RSUI)**	Literature review, interviews with patients and experienced clinicians	Test-retest, comparison of RSUI with other indicators of disease severity	VAS; SG	Multiplicative model	[49]
**Cancer**					
**European Organization for Research and Treatment of Cancer (EORTC-8D)**	Factorial analysis, Rasch analysis, expert opinion	Standardised Mean Response (SRM)	TTO	Multivariate regression model	[50]
**Quality of Life Questionnaire for Cancer 30 (QLQ-C30)**	Rasch model, basic psychometric criteria, clinical expert opinion	Not found	TTO	Random effects model	[44]
**Quality of Life Utility Measure—Core 10 Dimensions (QLU-C10D)**	Experts’ opinion; Confirmatory factor analysis (CFA); Rasch analysis; DIF; Patients’ opinion.	Not found	DCE	Conditional logistic regression	[51]
**Diabetes**					
**Diabetes Health Profile 3 and 5 dimensions (DHP-3D; DHP-5D)**	Exploratory factor analysis; consultation with professionals in the field; Rasch analysis.	Validation by professionals in the field	TTO	Generalized least squares with random effects	[52]
**Diabetes Utility Index (DUI)**	Non available	Comparison with other tools	VAS; SG	Simple linear regression model	[53]
**Sexuality/fertility**					
**International Index of Erectile Function (IIEF)**	Non available	Consistency of IIEF ordinal structure	TTO	Non available	[54]
**Labour and Delivery Index (LADY-X)**	Interviews with patients; Experts’ opinion.	Not found	DCE	Panel mixed logit model (MMNL)	[55]
**Sexual quality of life questionnaire (SQOL-3D)**	Psychometric criteria	Not found	TTO; DCE; Ranking	Ordinary least squares and random effects model; Ordered logit	[56]
**Bladder**					
**King’s Health Questionnaire (KHQ)**	Relevance of quality of life, percentage of items completed, face and construct validity of items, score distribution and responsiveness.	Not found	SG	Random effects models	[57]
**Overactive Bladder 5 dimensions (OAB-5D)**	Factorial analysis; Rasch analysis	Standardised response mean (SRM) method	TTO	Ordinary least squares; random effects model “one-way error components”.	[17,58]
**Menopause/flushing**					
**Flushing Symptoms Questionnaire (FSQ)**	Rasch analysis	Not found	TTO	Ordinary least square	[59]
**Menopause specific health quality of life questionnaire**	Focus group sessions with patients, literature review, expert opinion, standard psychometric criteria	Test–retest reliability, face validity, construct validity and convergent validity.	TTO	Random effects models	[60]
**Musculoskeletal disorders**					
**Dupuytren’s contracture (DC)**	Non available	Not found	DCE	Conditional logit	[61]
**Health Assessment Questionnaire for arthritis (HAQ)**	Rasch model, basic psychometric criteria, clinical expert opinion	Not found	TTO	Random effects model	[44]
**Vision/glaucoma**					
**Glaucoma Utility Index (GUI)**	Review of existing instruments on vision and glaucoma; advice from experts in the field	Not found	DCE	Conditional logit regression model	[62]
**Visual Function Questionnaire–Utility Index (VFQ-UI)**	Rasch analysis, expert opinion.	Not found	TTO	Multivariate regression	[63]
**Digestive function**					
**Short Bowel Syndrome-specific quality of life scale (SBS-QoL)**	Factor analysis and item performance analysis, expert opinion	Not found	LT-TTO	Random effects model	[64]
**Prostate**					
**International prostate symptom score (IPSS)**	Factorial analysis	Not found	TTO	Non available	[65]

Note: VAS = Visual analogue scale; TTO = Time trade-off; SG = Standard gamble; DCE = Discrete choice experiment; DCEtto = Discrete choice experiment with duration; BWS = Best-worst scaling; LT-TTO = Lead time-time trade-off.

In terms of the elicitation methods used for specific instruments, the TTO was the most frequently used method. Indeed, about half (*n* = 16) of the 36 instruments concerned were valued by this method. Some studies exclusively used a DCE (*n* = 8), VAS (*n* = 1), or best worst scaling (BWS) (*n* = 1). A mixed method was preferred in six studies, three of which used VAS and SG, another used TTO and VAS. The last two remaining studies estimated models with preferences obtained from three elicitation methods namely TTO, DCE, and BWS on the one hand and TTO, DCE, and ranking on the other hand. Then, comparisons were made to figure out which method (model) allows a better prediction of health states scores.

In order to estimate the utility scores of the various remaining combinations, the authors used different models such as random effects models (*n* = 12), simple ordinary or generalized least squares (*n* = 6), multiplicative models (*n* = 2), conditional logit or maximum likelihood models (*n* = 9), and multivariate/multinomial models (*n* = 3). Most of these different models proved their validity by the consistency of the model judged through its specifications (e.g., R^2^, root mean square error, SRM, sign of the coefficients, significance of the coefficients, AIC, and BIC criteria) (*n* = 17). Six studies made comparisons either with other instruments or with scores obtained with a population other than the one used in the initial study. One study used the Hausman test and Ljung-Box statistics, and another used the likelihood ratio (LR) test. Seven studies did not provide information on how the algorithm was validated.

## 4. Discussion

This work addressed the main steps in the development of preference-based measurement instrument for QALY calculation. The development of new instruments or the modification of existing ones requires an understanding of the different phases involved in the development of measurement tools. These phases are generally development, validation, and measurement. In this systematic review, 46 studies were selected, tracing the development of 48 preference-based instruments for use in economic evaluations. Among these instruments, 25 corresponded to improvements in existing instruments and 23 were developed de novo. Twelve instruments were generic and 36 were specific. The number of dimensions retained per instrument varied between 2 and 15 and the number of levels between 2 and 11. All generic instruments contained one or more dimensions related to physical discomfort/pain. Almost all these instruments addressed mobility/ambulation, at the expense of other dimensions such as mental/cognitive function, well-being/happiness/satisfaction, or sadness/depression. Literature reviews, Rasch analysis, IRT, and expert opinion were mostly used to select the final dimensions and levels for the different tools.

Most of the instrument’s validation was done by test–retesting, comparisons with other instruments, and the SRM method. However, several authors did not mention the method used to validate the instruments. The conversion algorithms were mainly designed using random effects models and the most widely used elicitation method was TTO. The study of the specifications of the different models and a comparison of the results obtained with those of different instruments or subsamples mainly allowed the final model to be retained. More than three-quarters (85%) of the studies provided details on the number of participants in the elicitation phase and almost all of those (95%) provided information on the characteristics of the participants.

At the time of study selection, rigor in methodology or the amount of information available was not a criterion for inclusion. For example, during data extraction, several studies did not provide information on important aspects of the tool development process such as the sampling strategy or the method of recruiting participant samples. In view of these aspects, it seems likely that biases may remain in the measurement of the utilities or in the algorithms derived from this information. Moreover, only 41% of the studies claimed to have used a representative sample of the target population in their work. This raises the question of the external validity of the various instruments.

Therefore, additional steps could be taken to ensure the operationality of the instrument or to provide a confidence interval for the results obtained. Sensitivity analysis is one such step. It is defined as a method to identify how different sources of uncertainty in the model (algorithm) can affect the value of the result obtained [66]. It thus makes it possible to account for the degree of stability or variability of the result provided. However, of all the studies selected, few were listed as having performed a sensitivity analysis (*n* = 3).

Nevertheless, the average quality of the studies constituting this review is acceptable and allows a clear description of the process used. Table 5 presents the quality of the different studies with regard to the COSMIN grid, which allows an evaluation of the quality of the studies according to different criteria (e.g., content validity, consistency, and reliability of the tool). Four levels of response are allowed, ranging from “very good” to “inadequate” depending on the criteria assessed. Table 5 provides the proportion of responses for each possible level of response and for the different criteria in the grid. On average, 56% of the various criteria assessed were rated as “very good” and 37% were rated as “doubtful or undetermined”. Only 6% of the criteria were rated, on average, as “inadequate”.

High diversity has been noticed in the instruments identified in this systematic review. This is the proof of an increasing interest in the field of health economics, especially in utility measurement. This provides researchers and policy makers diversified tools to appropriately assess programs and efficiently proceed to resources allocation. However, the variety of tools may also cause some concerns in programs comparison. It has been showed that instruments used in the same field may yield to different scores due to differences in their descriptive systems (i.e., domains of health covered) or valuation techniques used [16,67]. For example, Richardson et al. [16] found that individual with visual disturbances and hearing impairments had a score of 0.14 with the HUI 3 and 0.8 with the EQ-5D. Thus, a program that would allow a full recovery for this patient (i.e., utility of 1) would record a utility gain 4.3 times greater using the HUI 3 rather than the EQ-5D. This would mean that, for this program, the use of the EQ-5D would have the same effect than an increase of 4.3 times the cost on the cost/QALY ratio. To resolve these issues, guidelines as to which instrument to use in the estimation of QALY have been elaborated in some countries [67]. Decision makers must then be aware of pros and cons of each instrument to be able to select the most adequate one for their needs. This selection is nevertheless quite difficult to enforce in practice. In order to compensate this difficulty and provide important additional information on accuracy and reliability of results, Feeny et al. [68] recommend to use two or more instruments in studies.

Most of instruments in this systematic review are specific (*n* = 36). This shows the recent emphasis given to the development and use of specific instruments over generic instruments. Indeed, the development of most of the specific instruments discussed in this work is justified by the lack of sensitivity or psychometrical invalidity of generic instruments in the field concerned. From this point of view, it would be important to have a specific instrument for each field requiring it for a better measurement of preferences and a better allocation of resources. The specific instruments identified in this study cover only a small part of the many areas of health care, consequently, there is a long way to go to allow more domains to have a specific instrument.

In addition, the question of the target population in the process of developing the various specific instruments arises. For institutions, such as the US Public Health, it is important to use a representative sample of the general population if the assessments obtained are “informed, unbiased and competent” [69,70]. National Institute of Care for Health and Care Excellence (NICE) also advocates the argument that in a publicly funded health care system, the purpose of economic evaluation is not to make decisions at the individual patient level but to allow policies that serve the interests of society as a whole to emerge [69,71]. However, the use of utility values from the general population becomes problematic if these values differ substantially from those of patients. For this reason, several authors believe that patients should be directly addressed in the elicitation of preference scores [69,72,73]. It is nevertheless noted that nearly two thirds of the specific instruments present in this work used the preferences of individuals from the general population only (*n* = 26).

## 5. Conclusions

This systematic review on the development of preference-based instruments identified the various stages required to develop an instrument to measure QALY. This work thus provides a better understanding of the process of developing preference-based instruments for QALY calculation. Most of the studies that have focused on the development of specific instruments have been done because of the verified inadequacy of generic instruments in some areas. A great diversity was observed in the different methods used in the different stages of the development of the instruments. Rasch analysis, TTO, and random effects models were predominantly used in instrument development and measurement. Noting the high variability among studies in the process of developing the instruments included in this review, it would be very helpful to have a standardized method for the development of preferences-based instruments like what has been done for the experiment design of DCEs [74].

## Figures and Tables

**Figure 1 ijerph-18-04428-f001:**
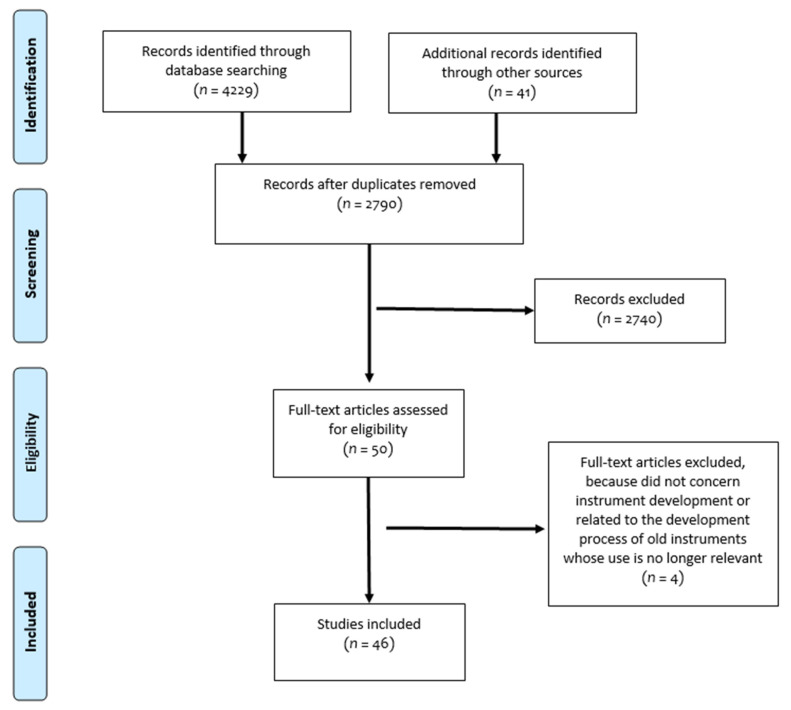
PRISMA flowchart, 18 June 2020.

**Table 1 ijerph-18-04428-t001:** Dimensions and levels selected in the generic tools.

	15D	AQoL-7D	AQoL-8D	CAT-5D-QOL	CORE-6D	EQ-5D-5L	HUI2	HUI3	PROMIS-29	QWB-SA	ReQoL-UI	SF-6Dv2
**Physical Domain**												
Vision	X	X	X				X	X		X		
Hearing/Listening	X	X	X				X	X		X		
Speech/Communication	X	X	X				X	X		X		
Breathing	X									X		
Eating/Nutrition	X	X	X							X		
Excretion	X									X		
Sleep	X		X						X	X		
Physical discomfort/Pain	X	X	X	X	X	X	X	X	X	X	X	X
Usual/Daily Activities	X	X	X	X	X	X			X	X		X
Self-care		X	X			X	X			X	X	X
Mobility/Ambulation	X	X	X	X		X	X	X	X	X	X	X
Dexterity/Handling				X				X		X		
Fertility							X					
**Mental Domain**												
Autonomy/Control/Dependence		X	X		X					X		
Adaptation/Coping		X	X								X	
Feeling of burden to other(s)			X									
Vitality/Energy	X	X	X						X	X		X
Mental/cognitive function	X	X					X	X		X		
Anxiety/Distress	X	X	X			X	X		X	X		X
Sadness/Depression	X	X	X			X	X		X	X		
Calm/Agitation/Irritability		X	X				X			X		
Anger			X				X			X		
Well-being/Happiness/Satisfaction			X	X			X	X	X		X	
Self-confidence/esteem			X						X		X	
Loneliness					X					X	X	
Enthusiasm/Pleasure			X					X		X	X	
Terror/Panic/Fear					X				X			
Humiliation/Shame					X							
Suicidal idea					X						X	
**Social Domain**												
Personal/Close relationship		X	X						X	X		X
Social inclusion/Connectedness		X	X						X			
**Other Domain**												
Appearance (deformity, weight, skin)										X		
Sexual activity	X									X		
Number of dimensions (items)	15	7 (26)	8 (35)	5 (25)	6	5	7	8	8 (29)	5 (77)	2 (7)	6
Number of levels by dimensions	5	4, 5, 6, 7	4, 5, 6	4	3	5	3, 4, 5	5, 6	5, 11	2, 4, 5	5	5, 6

**Table 2 ijerph-18-04428-t002:** Methods used for the development of generic instruments.

Instruments	Method of Choice for Dimensions and Levels	Validation Method	Elicitation Method	Model Used	References
15 dimensions (15D)	Factor analyses; patient surveys; instrument user feedback.	Multimethod multivariate matrices based on empirical measurements of the dimensions of 15D, NHP, SF-20 and EQ-5D.	VAS	Additive model	[20]
Assessment of Quality of Life (AQoL)-7D	Literature review and focus group; factor analysis; structural equation modelling; logical considerations.	Not found	TTO	Multiplicative regression model	[16]
Assessment of Quality of Life-8 (AQoL-8D)	Iterative process of entering and removing potential items in the AQoL model until all possible combinations are analyzed.	Loevinger H (homogeneity)	TTO	Multivariate linear regression	[10]
Computerized adaptative testing quality of life 5 dimensions (CAT-5D-QOL)	IRT	Not found	SG	Multiplicative regression model	[21]
Clinical Outcomes in Routine Evaluation 6 dimensions (CORE-6D)	Rasch analysis	Not found	TTO	Additive model	[2]
EuroQol 5 dimensions (EQ-5D-5L)	Literature review	Patients were asked to assess the interpretability and plausibility of the instrument.	VAS	Non applicable	[12]
Health Utilities Index 2 and 3 (HUI2-HUI3)	General population survey: the importance the public places on each attribute was considered.	Not found	VAS; SG	Multiattribute multiplicative model	[22]
Patient-Reported Outcomes Measurement Information System—29 (PROMIS-29 v2.0)	Item response theory; Factor (exploratory factor and confirmatory) analyses	Comparison with other instruments.	DCE	Relativity model	[23]
Quality of Well Being Self-Administered (QWB-SA)	Inputs from the QWB.	Test–retest; test the impact of the administration mode on total scores.	VAS	Additive model	[15]
Recovering Quality of Life utility index (ReQoL-UI).	Literature review, interviews, factor analyses and IRT	Not found	TTO	Random effects models	[24]
Short-Form 6-dimension (SF-6Dv2)	Exploratory and confirmatory factor analyses; Rasch analysis; literature review; expert opinion.	DIF on sub-samples.	DCEtto	Conditional Logit	[7,25]

Note: VAS = Visual analogue scale; TTO = Time trade-off; SG = Standard gamble; DCE = Discrete choice experiment; DCEtto = Discrete choice experiment with duration; IRT = Item Response Theory; NHP = Nottingham Health Profile; DIF = Differential item functioning.

**Table 3 ijerph-18-04428-t003:** Dimensions and levels selected in specific instruments.

Instruments	Number of Dimensions/Items	Nature of Dimensions	Number of Levels Per Dimension/Item
**Social care and dependency**			
Aberrant Behaviour Checklist Utility Index (ABC-UI)	7	Mood; Distractible; Aggressive; Impulsive; Speech; Social; Movements.	3
Adult Social Care Outcomes Toolkit (ASCOT)	8	Personal cleanliness and comfort; Accommodation cleanliness and comfort; Food and drink; Safety; Social participation and involvement; Occupation; Control over daily life; Dignity.	4
Carer Quality of Life—7 Dimensions (CarerQol-7D)	7	Fulfilment; Relational problems; Mental health problems; Problems with combining daily activities; Financial problems; Social support; Physical health problems of caregiving	3
Dependency 6 dimensions (DEP-6D)	6	Eat; Incontinence; Personal care; Mobility; Housework and Cognition/mental problems.	3, 4
Impact of Weight on Quality of Life—Lite (IWQOL-Lite)	8	Problems doing usual daily activities; Physical symptoms; Worrying about health; Low self-esteem; Sexual problems; Problems moving around or sitting in public places; Teasing or discrimination by others; Problems doing your job or getting recognition at work.	3
Index of capability for older people (ICECAP-O)	5	Attachment; Security; Role; Enjoyment and control.	4
Older Persons Utility Scale (OPUS)	5	Food and nutrition; Personal care; Safety; Social participation and involvement; Control over daily living.	3
**Neurological disorders**			
Alzheimer’s disease (AD-5D)	5	Interpersonal environment; Physical; Self-functioning; Memory; Mood.	4
Amyotrophic Lateral Sclerosis Utility Index (ALSUI)	4	Speech and swallowing; Eating; Dressing and bathing; Leg function and Respiratory function.	5, 6
Cerebral palsy-specific 6 dimensions (CP-6D)	6	Social well-being and acceptance; Physical health; Communication; Pain and discomfort; Manual ability; Sleep.	5
Epilepsy-specific preference-based measure (NEWQOL-6D)	6	Worry about attacks; Depression; Memory; Concentration; Stigma; control.	4
Multiple Sclerosis Impact Scale 29 (MSIS-29)	8	Problems with your balance; Being clumsy; Limitations in your social and leisure activities at home; Difficulties using your hands in everyday tasks; Having to cut down the amount of time you spent on work or other daily activities; Feeling mentally fatigued; Feeling irritable; impatient or short tempered; Problems concentrating.	4
Prototype Preference-Based MS Index (P-PBMSI)	5	Walking; Fatigue; Cognition; Mood; Work.	3
**Respiratory problems**			
Asthma Quality of Life (AQL-5D)	5	Concern; Short of breath; Weather and pollution; Sleep; Activities.	5
Cambridge Pulmonary Hypertension Outcome Review (CAMPHOR)	4	Social activities; Travelling; Dependence and Communication.	2, 3
Chronic obstructive pulmonary disease (COPD)	3	COPD; Non-serious exacerbations; Serious exacerbations.	3
Rhinitis Symptom Utility Index (RSUI)	5	Stuffy/blocked nose; Runny nose; Sneezing; Itchy/watery eyes and Itching nose/throat.	10
**Cancer**			
European Organization for Research and Treatment of Cancer (EORTC-8D)	8	Physical functioning; Role functioning; Social functioning; Emotional functioning; Pain; Fatigue and Sleep disturbance; Nausea; Constipation and diarrhoea.	4, 5
Quality of Life Questionnaire for Cancer 30 (QLQ-C30)	8	Trouble taking a long walk; Limited in doing either your work or other daily activities; Have you had pain; Have you felt nauseated; Were you tired; Difficulty in concentrating on things; Did you worry; Has your physical condition or medical treatment interfered with your social activities.	4, 7
Quality of Life Utility Measure—Core 10 Dimensions (QLU-C10D)	10	Physical functioning; Role functioning; Social functioning; Emotional functioning; Pain; Fatigue; Sleep; Appetite; Nausea; Bowel problems.	4
**Diabetes**			
Diabetes Health Profile 3 (DHP-3D)	3	Mood; Social limitations; Eating.	4
Diabetes Health Profile 5 (DHP-5D)	5	Mood; Social limitations; Eating; Hypoglycaemic attacks; Vitality.	4,5
Diabetes Utility Index (DUI)	5	Physical ability and energy; Relationships; Mood and feelings; Enjoyment of diet and Satisfaction with management of diabetes.	3, 4
**Sexuality/fertility**			
International Index of Erectile Function (IIEF)	2	Ability to Attain and maintain an erection sufficient for satisfactory sexual performance.	5
Labour and Delivery Index (LADY-X)	7	Availability of competent professionals; The information provided; Professionals’ responses to needs; Professionals’ emotional support; Feelings of safety; Concerns about the child’s condition; Duration until first contact with child.	3
Sexual quality of life questionnaire (SQOL-3D)	3	Sexual performance; Sexual relationship and Sexual anxiety.	4
**Bladder**			
King’s Health Questionnaire (KHQ)	5	Role limitation; Physical limitations; Social limitations/family life; Emotions; and Sleep/energy.	4
Overactive Bladder 5 dimensions (OAB-5D)	5	Urge; Urine loss; Sleep; Coping; Concern.	5
**Menopause/flushing**			
Flushing Symptoms Questionnaire (FSQ)	5	Redness of skin; Warmth; Tingling; Itching; Sleep difficulty	4, 5
Menopause specific health quality of life questionnaire	7	Hot flushes; Aching joints/muscles; Anxious/frightened feelings; Breast tenderness; Bleeding; Vaginal dryness and Undesirable androgenic signs.	3, 5
**Musculoskeletal disorders**			
Dupuytren’s contracture (DC)	8	Joint #1: index finger, PIP joint; Joint #2: index finger, MCP joint; Joint #3: middle finger, PIP joint; Joint #4: middle finger, MCP joint; Joint #5: ring finger, PIP joint; Joint #6: ring finger, MCP joint; Joint #7: little finger, PIP joint; Joint #8: little finger, MCP joint.	3
Health Assessment Questionnaire for arthritis (HAQ)	5	Stand up from a straight chair; Walk outdoors on flat ground; Get on / off toilet; Reach and get down a 5-pound object (such as a bag of sugar) from just above your head; Open car doors.	4
**Vision/glaucoma**			
Glaucoma Utility Index (GUI)	6	Central and near vision; Lighting and glare; Mobility; Activities of daily living; Eye discomfort; Other effects of glaucoma and its’ treatment	4
Visual Function Questionnaire–Utility Index (VFQ-UI)	6	Near vision activities; Distance vision activities; Vision-specific social functioning; Role difficulties; Dependency; and Mental health.	5
**Digestive function**			
Short Bowel Syndrome-specific quality of life scale (SBS-QoL)	6	Diet; Eating and drinking habits; Diarrhoea; Fatigue/weakness; Mobility and self-care/everyday activities; Leisure activities/social life; Emotional life.	2
**Prostate**			
International prostate symptom score (IPSS)	2	Obstructive symptoms; Irritative symptoms.	3

**Table 5 ijerph-18-04428-t005:** Analysis of the quality of studies using the COSMIN grid.

Authors	Very Good	Adequate	Doubtful/Undetermined	Inadequate
[2]	57.89%	-	36.84%	5.26%
[7]	57.89%	-	42.11%	-
[8]	42.11%	-	52.63%	5.26%
[9]	84.21%	-	10.53%	5.26%
[10]	57.89%	-	31.58%	10.53%
[12]	42.11%	-	47.37%	10.53%
[15]	47.37%	-	42.11%	10.53%
[16]	57.89%	-	36.84%	5.26%
[17]	52.63%	-	42.11%	5.26%
[20]	57.89%	-	36.84%	5.26%
[21]	57.89%	-	26.32%	15.79%
[22]	47.37%	-	47.37%	5.26%
[23]	78.95%	-	21.05%	-
[24]	57.89%	-	36.84%	5.26%
[25]	42.11%	-	31.58%	-
[35]	63.16%	-	36.84%	-
[36]	52.63%	-	47.37%	-
[37]	63.16%	-	26.32%	10.53%
[38]	63.16%	-	36.84%	-
[39]	57.89%	-	42.11%	-
[40]	57.89%	-	42.11%	-
[41]	47.37%	-	42.11%	5.26%
[42]	57.89%	-	36.84%	5.26%
[43]	36.84%	5.26%	52.63%	5.26%
[44]	57.89%	-	26.32%	15.79%
[45]	94.74%	-	5.26%	-
[46]	57.89%	-	36.84%	5.26%
[47]	63.16%	-	31.58%	5.26%
[48]	52.63%	-	42.11%	5.26%
[49]	36.84%	-	42.11%	21.05%
[50]	57.89%	-	31.58%	10.53%
[51]	73.68%	-	26.32%	-
[52]	57.89%	-	36.84%	5.26%
[53]	63.16%	-	31.58%	5.26%
[54]	52.63%	-	36.84%	10.53%
[55]	63.16%	-	36.84%	-
[56]	47.37%	-	47.37%	5.26%
[57]	63.16%	-	31.58%	5.26%
[58]	52.63%	-	42.11%	5.26%
[59]	57.89%	-	36.84%	5.26%
[60]	57.89%	-	36.84%	5.26%
[61]	42.11%	-	57.89%	-
[62]	57.89%	-	42.11%	-
[63]	52.63%	-	36.84%	10.53%
[64]	42.11%	-	47.37%	10.53%
[65]	36.84%	-	52.63%	10.53%

## Data Availability

Not applicable.

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
