# Peer review of "Dimensions Used in Instruments for QALY Calculation: A Systematic Review"

_ijerph, 2021, doi:10.3390/ijerph18094428_

Round 1

Reviewer 1 Report

The study is a systematic review of instruments to measure quality of life. Overall, the study is relevant and well conducted. 

I only have a few minor comments. In the introduction section, I think that the need for this study is not well argued.

I do not see the relevance of the paragraph on COVID-19. It should be removed. Just because the study was conducted during the pandemic  does not mean that such a context needs to be mentioned and impacts the study.

In the methods, for the search strategy (use this terminology rather than "research strategy"), it is mentioned that the "search ended on June 18, 2020". Does that refer to the date at which the search was conducted? The authors should instead clearly state the period covered in their search. Was there a limit to how far back the search went? And if so, what was the rationale?

In the results section, there is a mention of "as per protocol". Was there a protocol published? It does not seem to be the case so the reference should be instead to the methods section.

 In the discussion section, I think that the benefits and drawbacks of having such a diversity of instruments needs to be explained further. Also, it would be interesting to have more information on whether and the extent to which different instruments yield different QALY for a same population.

Author Response

Reviewer 1

The study is a systematic review of instruments to measure quality of life. Overall, the study is relevant and well conducted. 

I only have a few minor comments. In the introduction section, I think that the need for this study is not well argued.

This point has been considered (lines 19-22 of the introduction).

I do not see the relevance of the paragraph on COVID-19. It should be removed. Just because the study was conducted during the pandemic does not mean that such a context needs to be mentioned and impacts the study.

The paragraph has been deleted. Thanks for the suggestion.

In the methods, for the search strategy (use this terminology rather than "research strategy"), it is mentioned that the "search ended on June 18, 2020". Does that refer to the date at which the search was conducted? The authors should instead clearly state the period covered in their search. Was there a limit to how far back the search went? And if so, what was the rationale?

The search process started on May 4, 2020 and ended on June 18, 2020. A final check was performed on June 18, 2020 to consider potential new publications. There was no limit for how far back the search went. Consequently, the reference period was “up to June 18, 2020.”

The manuscript has been changed accordingly where appropriate.

In the results section, there is a mention of "as per protocol". Was there a protocol published? It does not seem to be the case so the reference should be instead to the methods section.

The protocol is not published, the reference has been modified accordingly to the reviewer suggestion (lines 8-9 of the selection of studies). The protocol was established to allow consistency between authors in the search strategy and the selection of studies. This is a two-pages document written in French for internal use.

In the discussion section, I think that the benefits and drawbacks of having such a diversity of instruments needs to be explained further. Also, it would be interesting to have more information on whether and the extent to which different instruments yield different QALY for a same population.

Great proposition! It has been added in the discussion section (lines 59-66 of the discussion).

Reviewer 2 Report

The topic of the review is of great interest for researchers and policy makers given the current emphasis on patients’ perspectives (without need to mention the coronavirus epidemic).

I am puzzled however by the use of the acronym QALY by the authors and I think there might be a misunderstanding and that they mean health state utility.

A QALY is the results of a health-related utility measure multiplied by a time period. The authors seem to make a confusion between the instruments used to elicit utility measures and the QALY, which makes the manuscript quite difficult to read.

I think the authors meant to propose a review of health-related quality of life instruments (both generic and specific) that are used to obtain health state utility values, so the term QALY in the title and in the text is used is inappropriately because here is no QALY instrument per se.

In the description of their objectives, the authors are not clear about whether they study the developments of instruments that were a priori designed to determine utilities or instruments that were later used, through mapping algorithms, to obtain utilities and calculated QALYs. I think there is an important difference here that the authors do not seem to take into account. The authors write in the results: ‘This review thus considered 46 studies dealing with the development of 48 preference-based instruments for the purpose of QALY calculation’. I do not think that it is correct, some of the intstruments were never meant initially for QALY calculation.

A more rigorous approach would be to compare the development  of instuments that were initially meant to be used in QALY calculations to instruments that were not.

The tables are well done and useful, they could be recycled easily with a revised.

In the discussion, I disagree with the sentence ‘This work also highlights the magnitude of the task to perform in terms of developing specific instruments for the no less important areas not covered by the studies included in this review’. If the review is to ensure that all disease areas are covered by disease-specific instruments, it is a completely different study than the one presented here.  Disease-specific instrument are by definition ill suited to the production utility values, it is therefore unclear what the authors meant. The ‘the verified inadequacy of generic instruments ‘ concerns sensitivity to change for example, but mapping specific instruments into utilities also has limitations (Louise Longworth, Donna Rowen,Mapping to Obtain EQ-5D Utility Values for Use in NICE Health Technology Assessments,Value in Health,Volume 16, Issue 1,2013,Pages 202-210).

The authors have cited several publication by John Brazier, but I think not one of the most relevant for their work: J.E. Brazier, Y. Yang, A. Tsuchiya, et al. A review of studies mapping (or cross walking) non-preference based measures of health to generic preference-based measures. Eur J Health Econ, 11 (2010), pp. 215-225

Minor comments:

Methods:

what was the start date of the search?

Was the review registered in Prospero?

Author Response

Reviewer 2

The topic of the review is of great interest for researchers and policy makers given the current emphasis on patients’ perspectives (without need to mention the coronavirus epidemic).

The paragraph relative to the COVID-19 pandemic has been deleted.

I am puzzled however by the use of the acronym QALY by the authors and I think there might be a misunderstanding and that they mean health state utility.

A QALY is the results of a health-related utility measure multiplied by a time period. The authors seem to make a confusion between the instruments used to elicit utility measures and the QALY, which makes the manuscript quite difficult to read.

We understand the confusion it seems to create. We changed it for “instrument for QALY calculation” where appropriate.

I think the authors meant to propose a review of health-related quality of life instruments (both generic and specific) that are used to obtain health state utility values, so the term QALY in the title and in the text is used is inappropriately because here is no QALY instrument per se.

The appellation “QALY instrument” has been changed in both title and manuscript.

In the description of their objectives, the authors are not clear about whether they study the developments of instruments that were a priori designed to determine utilities or instruments that were later used, through mapping algorithms, to obtain utilities and calculated QALYs. I think there is an important difference here that the authors do not seem to take into account. The authors write in the results: ‘This review thus considered 46 studies dealing with the development of 48 preference-based instruments for the purpose of QALY calculation’. I do not think that it is correct, some of the instruments were never meant initially for QALY calculation.

The difference mentioned by the reviewer is truly important.

As indicated in the review (see Tables 2 and 4), except one instrument (AD-5D) for which it is planned to develop a value set in the future, all the instruments considered have provided the elicitation method used to obtain the value set and almost all of them have detailed the model used for the estimation. Even if some instruments were initially non-preference-based, they have been adapted in this purpose and the “modified” preference-based version were subsequently used in elicitation studies to get a value set. So, none of the instrument identified in this review were selected if the value set was the result of the mapping function. As mentioned in the paper, we were interested in the 3 phases of development process: development, validation, and measure. All the instruments in the review have their own value set.  

The structure of the sentence has been modified to avoid further confusion (lines 11-12 of the selection studies).

A more rigorous approach would be to compare the development of instruments that were initially meant to be used in QALY calculations to instruments that were not.

In the inclusion criteria we were interested by the nature of the instruments (i.e., selected if it is a preference-based instrument). Even if the original version wasn’t preference-based, we considered the “modified” version that has a value set and is usable for QALY calculation. We then investigated the development process for all instruments considered in this review.

The suggestion of the reviewer is a nice perspective to consider for further work.   

The tables are well done and useful, they could be recycled easily with a revised.

Thanks.

In the discussion, I disagree with the sentence ‘This work also highlights the magnitude of the task to perform in terms of developing specific instruments for the no less important areas not covered by the studies included in this review’. If the review is to ensure that all disease areas are covered by disease-specific instruments, it is a completely different study than the one presented here.  Disease-specific instrument are by definition ill suited to the production utility values, it is therefore unclear what the authors meant. The ‘the verified inadequacy of generic instruments ‘ concerns sensitivity to change for example, but mapping specific instruments into utilities also has limitations (Louise Longworth, Donna Rowen,Mapping to Obtain EQ-5D Utility Values for Use in NICE Health Technology Assessments,Value in Health,Volume 16, Issue 1,2013,Pages 202-210).

In this section, we wanted, on one hand, to highlight the increase in the development of specific instruments due to inadequacy of generic instruments in these fields. On the other hand, we tried to show the amount of work to do to allow each domain to have a specific instrument.

We have modified the sentnce to avoid further confusions (lines 67-69 of the Discussion).

The authors have cited several publication by John Brazier, but I think not one of the most relevant for their work: J.E. Brazier, Y. Yang, A. Tsuchiya, et al. A review of studies mapping (or cross walking) non-preference based measures of health to generic preference-based measures. Eur J Health Econ, 11 (2010), pp. 215-225

Thanks for the paper suggestion. The reference is now added.

Minor comments:

Methods:

what was the start date of the search?

The search process started on May 4, 2020, and the reference period was up to June 18, 2020, that is to say that there was no specific starting date for the search, it was a search among all publications in the database used up to June 18, 2020.

Was the review registered in Prospero?

The review was not registered in Prospero.

Round 2

Reviewer 2 Report

The authors have revised their manuscript to answer most of my comments, there are however a few minor issues still pending:

To my comment on the selection of preference-based instruments, the authors replied/ ‘In the inclusion criteria we were interested by the nature of the instruments (i.e., selected if it is a preference-based instrument). Even if the original version wasn’t preference-based, we considered the “modified” version that has a value set and is usable for QALY calculation. ‘

This is fine, but then they need to edit the point 3.4 in the results section which limits the results to preference-based intruments.

In the Discussion, the sentence ‘However, the variety of tools may also cause some concerns in programs comparison. It has been

showed that instruments used in the same field may yield to different QALY due to differences in their descriptive systems (i.e., domains of health covered) or valuation techniques used’ uses ‘QALY’ incorrectly.  

The sentence ‘Decision makers must then be aware of pros and cons of each instrument to be able to select the most adequate one for their needs’ following the very good example with EQ5D and HUI is a bit slippery. There could be a sentence about the difficulty to enforce that choice and the fact that many investigators would use disease specific instruments and map them to get utilities.

The real issue is whether health-related utilities obtained through mapping exercises of disease-specific instruments are comparable and really allow health maximization under budget constraint.

This part of the Discussion is unclear because the authors do not have a consistent view of the purpose of an instrument used in QALY calculations. I agree with the statement in the abstract  that the objective is to perform  cost utility comparisons  (the important word here is comparisons, which I suppose are meant between medical specialties ). So the discussion should also be about how disease-specific instruments are able to allow those comparisons and whether the development of more disease-specific instruments to cover all disease areas is needed.

Author Response

Answers to reviewer 2

The authors have revised their manuscript to answer most of my comments, there are however a few minor issues still pending:

To my comment on the selection of preference-based instruments, the authors replied/ ‘In the inclusion criteria we were interested by the nature of the instruments (i.e., selected if it is a preference-based instrument). Even if the original version wasn’t preference-based, we considered the “modified” version that has a value set and is usable for QALY calculation. ‘

This is fine, but then they need to edit the point 3.4 in the results section which limits the results to preference-based intruments.

This is unclear what you mean by “edit” since this section refers to all instruments that are preference-based. Whether they were originally developed for this or “modified” subsequently to be preference-based. However, we added the following sentence in this section: “In addition, it should be noted that some of the preference-based tools were originally developed in that purpose while others were pre-existing instruments that were modified to be preference-based »

In the Discussion, the sentence ‘However, the variety of tools may also cause some concerns in programs comparison. It has been showed that instruments used in the same field may yield to different QALY due to differences in their descriptive systems (i.e., domains of health covered) or valuation techniques used’ uses ‘QALY’ incorrectly.  

The modification has been made (line 6 in the same paragraph).

The sentence ‘Decision makers must then be aware of pros and cons of each instrument to be able to select the most adequate one for their needs’ following the very good example with EQ5D and HUI is a bit slippery. There could be a sentence about the difficulty to enforce that choice and the fact that many investigators would use disease specific instruments and map them to get utilities.

Some explanations have been added to this section accordingly to this comment (lines 16-19 of the same paragraph).

The real issue is whether health-related utilities obtained through mapping exercises of disease-specific instruments are comparable and really allow health maximization under budget constraint.

The reviewer addresses a relevant point than can be of great interest in a future study. This review’s aim was to relate and document the development process of preference-based instruments in the more detailed and understandable way possible. Since we did not address “mapping” in this review, we think that to discuss on that point would be misleading for the readers.

This part of the Discussion is unclear because the authors do not have a consistent view of the purpose of an instrument used in QALY calculations. I agree with the statement in the abstract  that the objective is to perform  cost utility comparisons  (the important word here is comparisons, which I suppose are meant between medical specialties ). So the discussion should also be about how disease-specific instruments are able to allow those comparisons and whether the development of more disease-specific instruments to cover all disease areas is needed.

Comparisons can be done between programs from different specialities or with programs belonging to the same field. Generic instruments are known to allow comparisons between programs whether they belong to the same field or not. For specific instruments, because they were developed for a specific field, they don’t allow comparison between different medical fields, they just compare programs from the same field (see section 3.3). Since specific instruments are not able to compare programs from different fields, it will be interesting to have disease-specific instruments to cover as many as possible fields, and this for two reasons: 1- it will be possible to have an alternative for generic instruments and  better assess programs because specific instruments are known to be more responsiveness; 2- as stated by Feeny and al. (2019), specific instruments and generic instruments could be used together in studies to enhance the accuracy and reliability of results. Also, the possible need of more disease-specific instruments to cover all disease areas has already been addressed in the Discussion (penultimate paragraph of the Discussion).